# Interlayer donor-acceptor pair excitons in MoSe₂/WSe₂ moiré heterobilayer

Hongbing Cai[1,2], Abdullah Rasmita [1], Qinghai Tan [1], Jia-Min Lai[3,4], Ruihua He[5], Xiangbin Cai [1], Yan Zhao[1], Disheng Chen[1,2], Naizhou Wang [1], Zhao Mu[1], Zumeng Huang[1], Zhaowei Zhang [1], John J. H. Eng[1,6], Yuanda Liu[1,2], Yongzhi She[7], Nan Pan[7], Yansong Miao [5], Xiaoping Wang[7], Xiaogang Liu[8], Jun Zhang [3] ✉ & Weibo Gao [1,2,9] ✉

Localized interlayer excitons (LIXs) in two-dimensional moiré superlattices exhibit sharp and dense emission peaks, making them promising as highly tunable single-photon sources. However, the fundamental nature of these LIXs is still elusive. Here, we show the donor-acceptor pair (DAP) mechanism as one of the origins of these excitonic peaks. Numerical simulation results of the DAP model agree with the experimental photoluminescence spectra of LIX in the moiré MoSe₂/WSe₂ heterobilayer. In particular, we find that the emission energy-lifetime correlation and the nonmonotonic power dependence of the lifetime agree well with the DAP IX model. Our results provide insight into the physical mechanism of LIX formation in moiré heterostructures and pave new directions for engineering interlayer exciton properties in moiré superlattices.

The 2D transition metal dichalcogenide (TMD) bilayer is a promising platform for the study of strongly correlated electronic phenomena[1–7] and quantum optoelectronics applications. The moiré superlattice, which arises from the difference in lattice constant or a particular range of twist angle[8–10], can increase the effect of electron-electron interactions[11,12]. It also affects composite quasiparticles such as excitons (i.e., electron-hole pairs)[13–20], including interlayer excitons (IXs), where the electron and hole reside in different layers. The superlattice potential landscape confines IX in real space, resulting in a localized IX (LIX) with brightness that can be enhanced by placing the heterostructure on nanopillars[21]. Due to the localization effect and electron-exciton interaction, LIX is an excellent sensor for detecting strong electron-electron interactions[22–25]. Moreover, considering its large binding energy and nonzero out-of-plane electrical dipole[26,27], LIX could be used for

electrically tunable quantum emitter arrays[28] operating at elevated temperatures.

One of the puzzling aspects of LIX emission in TMD moiré heterobilayers is the sharp and dense peaks observed in its photoluminescence (PL) spectrum[20,21,28–31]. While magneto-optical spectroscopy measurements have shown that these peaks are related to band-edge excitonic states[20,28,29], the number of these states predicted from theoretical calculations is only 10 states[14,18] or up to 20 states if the triplet IX state brightening[32] is considered. This number is much smaller than the number of observed peaks (more than 50), indicating other possible mechanisms[33]. It is urgent to uncover this mechanism as it is the key component for correlated phase manipulation and single photon emitter engineering.

Here, by analyzing the spectra and lifetimes of LIXs, we show that most of the LIX peaks in molybdenum diselenide (MoSe₂)/tungsten

[1]Division of Physics and Applied Physics, School of Physical and Mathematical Sciences, Nanyang Technological University, Singapore 637371, Singapore. [2]The Photonics Institute and Centre for Disruptive Photonic Technologies, Nanyang Technological University, Singapore 637371, Singapore. [3]State Key Laboratory of Superlattices and Microstructures, Institute of Semiconductors, Chinese Academy of Sciences, Beijing 100083, China. [4]Center of Materials Science and Optoelectronics Engineering, University of Chinese Academy of Sciences, Beijing 100049, China. [5]School of Biological Sciences, Nanyang Technological University, Singapore 637551, Singapore. [6]Institute of Materials Research and Engineering, Agency for Science, Technology and Research, Singapore, Singapore. [7]Department of Physics, University of Science and Technology of China, Hefei Anhui 230026, China. [8]Department of Chemistry, National University of Singapore, Singapore 117543, Singapore. [9]Centre for Quantum Technologies, National University of Singapore, Singapore, Singapore. ✉e-mail: zhangjwill@semi.ac.cn; wbgao@ntu.edu.sg

diselenide (WSe$_2$) heterobilayer on nanopillars come from donor-acceptor pair (DAP) exciton recombination. The DAP exciton[34–37] is a bound pair of an electron trapped in the donor site and a hole trapped in the acceptor site with a narrow linewidth PL emission due to its localization (see Fig. 1a). Its emission energy depends on the donor-acceptor distance[34], resulting in multiple sharp PL peaks. The interlayer DAP exciton (DAP IX) emission can account for more than 75% of the experimentally observed peak positions.

Similar to the emission of localized defect exciton in TMD monolayer[38,39], the IX potential landscape created by the nanopillar-induced strain increases the coupling between the band-edge IX and localized DAP IX, enhancing the DAP IX emission (see Fig. 1b). By considering this coupling and the DAP exciton energy-lifetime relationship (i.e., lower energy DAP exciton has a longer lifetime[40]), we construct a DAP IX dynamic model, which fits well with the experimentally observed LIX energy-lifetime correlation. Additionally, we have experimentally observed a unique nonmonotonic power dependence of the energy-resolved lifetime, which agrees well with the DAP IX dynamic model. The agreement between the experimental and theoretical results is strong evidence that the DAP IX is the primary origin of multiple sharp PL peaks in moiré heterobilayers. Our results reveal a novel physical mechanism behind LIX emission and open a new strategy for LIX emission modulation.

## Results

### Device structure and PL intensity map

The sample structure used for the experiment is shown in Fig. 1c, and the optical microscope image of the sample is shown in Fig. 1d (see Supplementary Information Fig. S1a, b for the scanning electron microscope (SEM) image and see Methods for more details on the fabrication procedure). It consists of an hBN-encapsulated MoSe$_2$/WSe$_2$ heterostructure on a nanopillar array. Based on the second harmonic generation (SHG, Fig. 1e), the twist angle between the layers is close to $60 \pm 1°$, i.e., an AB-stacking configuration[41]. This assignment

of stacking configuration is also consistent with the g-factor measurement of the IX PL emission, showing a g-factor close to 16, expected for IX in AB stacked MoSe$_2$/WSe$_2$[20,42] (see Supplementary Information Fig. S2). As shown in Fig. 1f, the IX PL intensity map shows a significantly larger intensity for excitation at the nanopillar location than in the flat region, indicating that the LIX emission dominates the IX PL emission.

### PL spectrum of donor-acceptor pair IX

We next measured PL spectrum from the nanopillar locations. As shown in Fig. 2a–c, the LIX PL emission consists of sharp and dense peaks, with the linewidth in the order of 100 µeV (see Supplementary Information Fig. S3 for PL peaks Lorentzian fitting), in contrast with the broad PL spectrum of the WSe$_2$ and MoSe$_2$ intralayer exciton on a flat monolayer (Supplementary Information Fig. S4). The LIX peaks have energy close to the band-edge IX central energies (around 1.4 eV for unstrained AB-stacked MoSe$_2$/WSe$_2$[42,43]). PL spectra from other nanopillar locations also show the same behaviour (see Supplementary Information Fig. S5). We then extract the peak position and compare them with the peak position predicted by the DAP model, which can be approximated as[34]

$$E(R_m) = E_0 + \frac{\alpha}{R_m}, \tag{1}$$

where $E_0$ is the minimum DAP IX energy, $R_m$ is the $m^{th}$ distance between the donor and acceptor (with integer $m$ being the shell number), and $\alpha = \frac{e^2}{4\pi\varepsilon}$ where $e$ is the electron charge and $\varepsilon$ is the effective permittivity. The possible $R_m$ values are determined by the lattice constant and interlayer distance (see Fig. 2d for an illustration of $R_1$ and $R_2$). The predicted DAP IX peak positions are shown as red lines in Fig. 2a–c and plotted together with the experimentally obtained peak positions in Fig. 2e–g. More than 75% of the observed peaks match the predicted DAP IX peaks (Fig. 2h), indicating a good

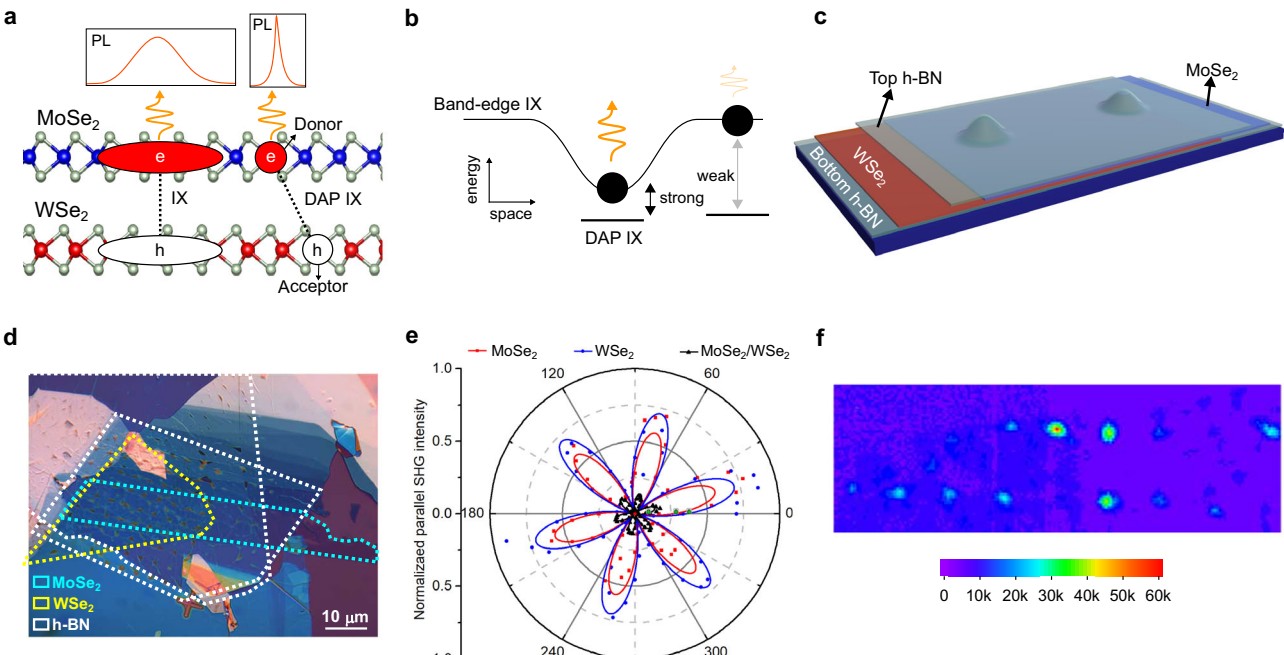

**Fig. 1 | Device structure and LIX emission. a** Illustration of DAP IX. Localized DAP IX has a sharper emission than unlocalized IX. **b** Enhancement of DAP IX emission by nanopillar. The strain-induced IX PL potential landscape enhances the coupling between band-edge IX and DAP IX, brightening the DAP IX emission. **c** Sample structure. The sample consists of an hBN-encapsulated MoSe$_2$/WSe$_2$ on nanopillars.

**d** Optical microscopy image of the heterostructures fabricated by the all-dry transfer stacking method. **e** Angle-dependent SHG intensity. The SHG shows that the twist angle is ~ $60 \pm 1°$. **f** IX integrated PL intensity map at 4.2 K under 765 nm (1.62 eV) laser excitation with an 850 long pass filter on the collection arm. Significant IX PL enhancement is observed at nanopillars, indicating LIX emission.

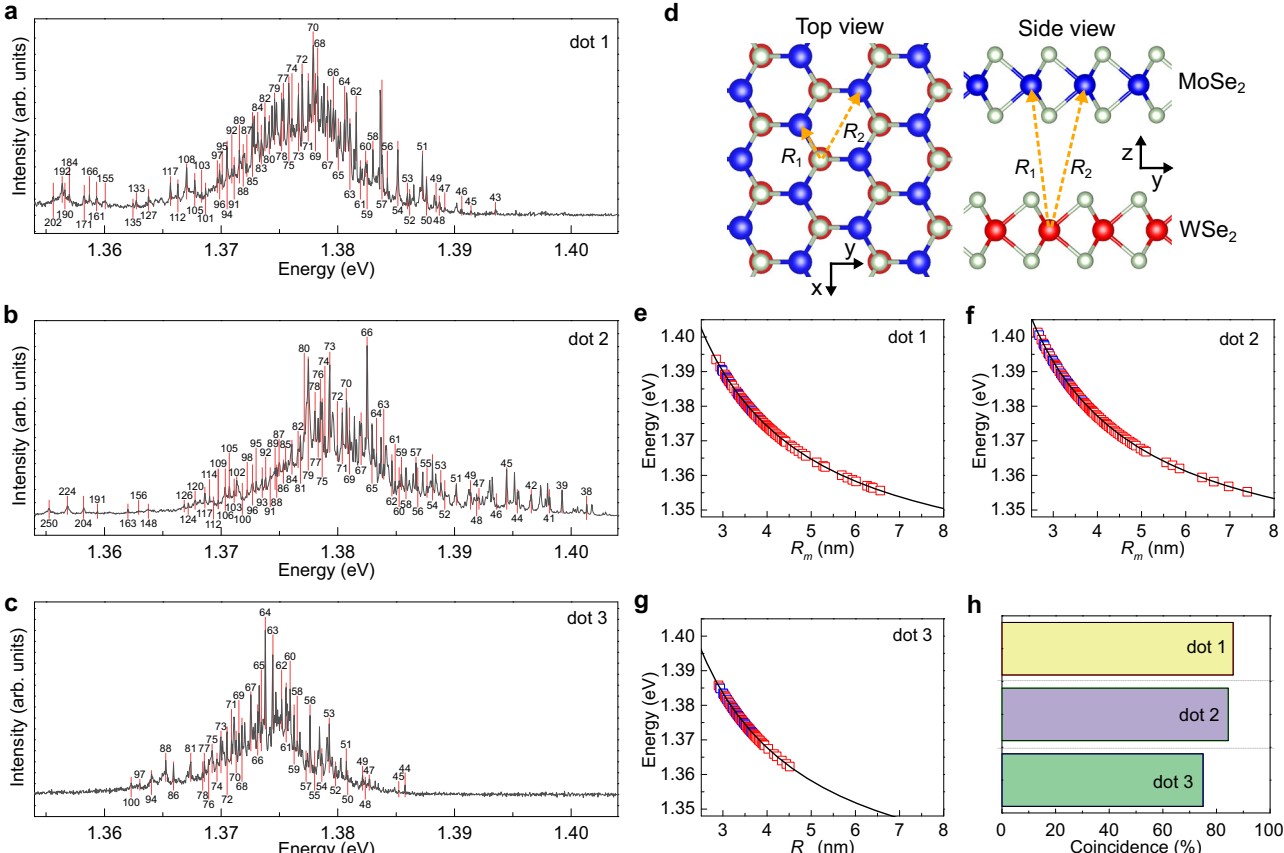

**Fig. 2 | DAP IX PL spectra. a–c** IX PL spectra. Solid black lines are the measured spectra. Red dash lines are the predicted DAP IX peaks, with the labels indicating the shell numbers. **d** Illustration of DAP IX in the van der Waals heterobilayer. $R_1$ and $R_2$ are the first two smallest distances between the donor-acceptor pair. **e–g** Calculated DAP IX peak positions (lines) vs. experimentally obtained peak positions (symbols). The red (blue) symbols are the peak positions that (do not) match the DAP IX peak positions. **h** DAP model goodness of fit. The coincidence, defined as the number of experimentally observed peaks matching the DAP IX peaks divided by the total number of experimentally observed peaks, represents the goodness of fit.

agreement between the DAP model and experimental results (see Supplementary Information Section 1 for more details on the PL peak analysis). More discussion on the donor-acceptor pair nature, including transmission electron microscopy (TEM) characterization, is presented in Supplementary Information Section 5.

We note that the PL signals from areas other than nanopillars, including the wrinkle and the flat areas, also consist of dense and narrow peaks, albeit with weaker PL intensity (see Supplementary Information Fig. S6). This result shows that the primary role of nanopillar-induced strain is to enhance PL emission (as in Fig. 1b) and not to create new emitters. More discussion on the role of nanopillar-induced strain is presented in Supplementary Information Sections 6 and 7.

From Fig. 2e–g, we found that the maximum donor-acceptor distance is ~8 nm, which agrees with the DAP IX model. In particular, due to the deep moiré superlattice potential in slightly twisted MoSe$_2$/WSe$_2$[44], it is unlikely that the donor and acceptor are located at two different moiré sites. Hence, the maximum donor-acceptor distance must be less than the moiré period. In our case of $60 \pm 1°$ twisted MoSe$_2$/WSe$_2$, the moiré period is ~20 nm[17], which is significantly larger than the maximum-donor acceptor distance, indicating the adequacy of the DAP IX model. This dependence on the moiré superlattice size also agrees with the PL spectra from the samples with different twist angles (see Supplementary Information Section 4). Moreover, this dependence also shows that most sharp peaks cannot be attributed to defect-bound exciton (i.e., exciton bound to a single defect), whose PL spectra should not show moiré superlattice dependence[33].

## Lifetime-energy correlation of DAP IX

Further evidence for the DAP model is obtained from energy- and time-resolved PL measurements of the LIXs. The IX PL spectrum time trace under pulsed 726-nm diode laser excitation shows no jittering behaviour (Fig. 3a), allowing a reliable time-resolved PL measurement of an emission peak. A homemade spectrometer (Fig. 3b) was used to filter the emission of a chosen peak for the time-resolved PL measurement (see Supplementary Information Fig. S7 for filtered PL emission spectra). Typical time-resolved PL of different emission peaks are shown in Fig. 3c. The experimental data fit well with the double exponential decay function, indicating that at least two different states are involved. We attribute these two states to DAP IX and band-edge IX states.

The extracted decay rate constants are plotted in Fig. 3d. Instead of a monotonous change, fast ($k_1$) and slow ($k_2$) decay rates reach maximum values at particular energy values. This behaviour cannot be explained using the usual spatially-localized exciton framework (e.g., defect-bound exciton), where a higher energy exciton is expected to decay faster due to the larger number of decay channels[45,46]. Instead, this observation can be explained by considering the dynamics of DAP IX and its coupling to the band-edge IX (see inset in Fig. 3d), where DAP IX decays faster than band-edge IX. For the case where $k_1 \gg k_2$, the decay rates can be approximated as

$$k_1 \approx k_{DAP} = k_{nr} + k_{r0} \exp\left(-\frac{2\alpha}{R_B(E - E_0)}\right), \quad (2)$$

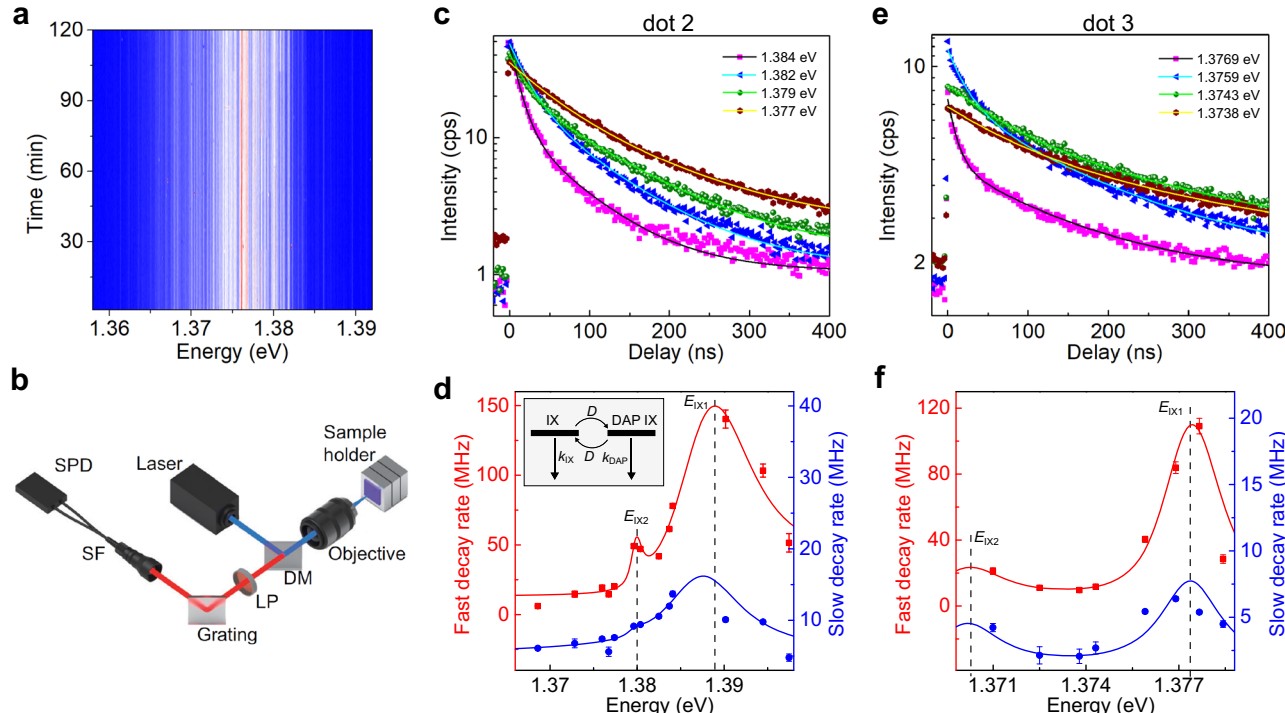

**Fig. 3 | Lifetime-energy correlation of LIX emission. a** IX PL spectrum time trace. **b** Experimental setup. A high transmittance grating was used to spatially separate emissions from different peaks. DM Dichroic mirror, LP Long pass filter, SF Single mode fibre, SPD superconducting single-photon detector. **c** Energy- and time-resolved PL of LIX emission. Data (shown as symbols) was obtained with excitation location at dot 2 (as in Fig. 1c) under a 6 nW excitation. The lines are double exponential decay fitting results. **d** Decay rates vs emission energy for dot 2. The lines are the fitting results using the DAP IX dynamic model. The dashed lines mark the central energies of band-edge IXs used for the fitting. Inset: DAP-band edge IX coupling model. The downward straight arrows represent decay rates, while the curved arrows represent coupling rates. **e, f** Like c and d but for dot 3 under a 2 nW excitation.

$$k_2 \approx k_{IX} + D, \tag{3}$$

where $k_{DAP}$ is the DAP IX total decay rate, $k_{nr}$ is the DAP IX non-radiative decay rate, $k_{r0}$ is the DAP IX maximum radiative decay rate, $R_B$ is the donor (acceptor) Bohr radius (whichever is largest), $k_{IX}$ is the band-edge IX decay rate, $D$ is the coupling rate between the band-edge and DAP IX, and other parameters are the same as in Eq. (1). Considering the coupling between the DAP and band-edge IX, the values of $R_B$ and $D$ depend on the energy difference between these two IX states. The coupling rate between the DAP and band-edge IX is maximum when their energy is the same. Due to this coupling, the DAP IX state is a combination of the uncoupled DAP IX and band-edge IX states. Since the band-edge carrier has a larger Bohr radius than the donor (acceptor) one, the coupling also increases the value of DAP IX's $R_B$. Hence, both $R_B$ and $D$ reach maximum values for the peak energy equal to the band-edge IX central energy, which results in the maximum values for both fast and slow decay rates according to Eqs. (2) and (3). As shown in Fig. 3d, there is an excellent agreement between this model and the data, further confirming the DAP IX mechanism (see Supplementary Information Section 2 for more details on the calculation of DAP IX energy-lifetime correlation). Two band-edge IX peaks were detected (labelled as $E_{IX1}$ and $E_{IX2}$), with ~7 meV peak separation. This peak separation value agrees with the IX trion binding energy in MoSe$_2$/WSe$_2$[23,47], indicating that these two peaks are neutral and charged band-edge IX peaks. Similar behaviour is also observed for other dots (see Fig. 3e, f for the results for dot 3) and for the PL emission from the flat area (Supplementary Information Fig. S8).

## Excitation power dependence of DAP IX

The dependence of the decay rate on the coupling between the DAP IX and band-edge IX suggests that the power dependence of the decay rates should also show a nonmonotonous trend. In general, IX decay rates increase with increasing power due to the increase in the non-radiative decay rate[48]. However, an increase in power also causes a power-induced blue shift in IX energy due to dipole-dipole repulsion[49,50], which can have different magnitudes for localized DAP IX and band-edge IX[31]. As illustrated in Fig. 4a, a DAP IX peak which is initially resonant with the band-edge IX at low power, becomes less resonant as power increases. The increased energy gap between the band-edge and DAP IX results in the decoupling between the two IX states, reducing the Bohr radius. Consequently, the fast decay rate decreases when the power is increased (see Eq. (2)). Further increasing the power does not result in significant decoupling, and the usual power-induced increase in decay rate is observed. Hence, in addition to the general density-dependent increase, decay rates, especially the fast decay rate, should exhibit nonmonotonous energy-dependent change.

To verify this, we measured the power dependence of the energy- and time-resolved PL for power from 2 nW to 40 nW. Power-dependent PL spectra (Supplementary Information Fig. S9) can be fitted with multiple Lorentzians, indicating that the emission in this power range is still dominated by LIX emission. Individual peak positions do not show a power-dependent blueshift, agreeable with previous reports[31]. On the other hand, the overall PL strength is blueshifted at higher power due to the band-edge IX blueshift. Similar behaviour is observed in the power-dependent PL spectrum from the flat area (Supplementary Information Fig. S10).

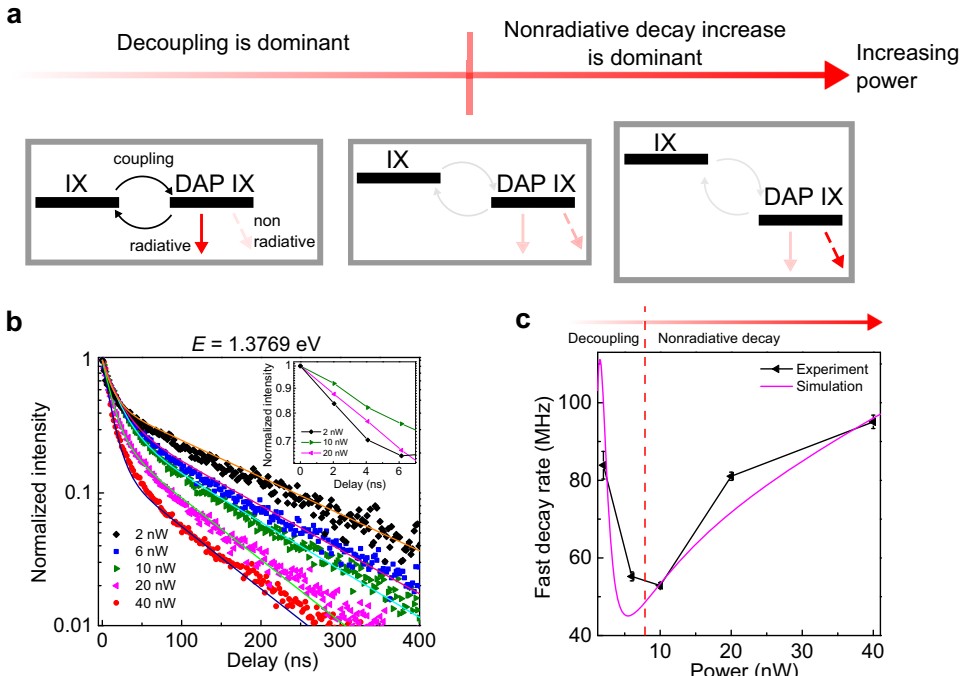

**Fig. 4 | Power dependence of the LIX lifetime. a** Schematic of power-dependent decay rate mechanism. The downward straight arrows represent decay rates with solid (dashed) lines indicating (non)radiative decay rates, and darker colour means faster decay rates. The curved arrows represent coupling rates. With increasing power, the nonradiative rate increases. Additionally, the energy spacing of the DAP-band edge IX increases, reducing the coupling and narrowing the Bohr radius, resulting in a decrease in the DAP IX radiative decay rate. **b** Normalized time-resolved photoluminescence of LIX with the same photon energy (1.3769 eV) under different laser powers. The data (shown as symbols) are obtained with the excitation location at dot 3 (as in Fig. 1c and Fig. 3e, f). The inset shows measured data for 2, 10, and 20 nW at the 0–7 ns time range, exhibiting the nonmonotonous power dependence of the fast decay rate. The lines are the results of double exponential decay fitting. **c** Experimental and simulated fast decay rate vs excitation power.

The measured power-dependent time-resolved PLs for the peak with an energy of 1.3769 eV are shown in Fig. 4b, together with the double exponential fitting. As predicted, the extracted fast ($k_1$) decay rate, shown in Fig. 4c, exhibits a nonmonotonous power dependence. We simulate power-dependent decay rates following the DAP IX dynamic model. Simulation results show similar trends to experimental results, supporting the DAP IX hypothesis. The simulated results for other peaks, including the fast and slow decay rates, are also consistent with the experimental result (see Supplementary Information Fig. S11 and Supplementary Information Section 3). We note that the change in the energy difference between band-edge and localized DAP IX is also apparent from the power-dependent lifetime-energy correlation (see Supplementary Information Fig. S12).

## Discussion

In summary, we have theoretically proposed and experimentally confirmed a novel DAP IX mechanism as the origin of the dense and sharp IX emissions from MoSe₂/WSe₂ on nanopillars, besides other mechanisms that could take place. Our experimental results, including energy- and time-resolved PL measurements and the lifetime's power dependence, provide strong evidence for the DAP IX model. Our work renders novel insight into the nature of LIX emission in 2D heterostructures and highlights the interplay between the disorder (represented by the DAP IX) and superlattice potential (represented by the band-edge IX) in the correlated moiré material. In terms of application, the multipeak nature and the electric field tunability of the LIX emission (i.e., due to its out-of-plane dipole direction) show that it is a versatile quantum emitter platform. When a smaller number of peaks is needed, it is possible to reduce them using a smaller sample, as shown in a recent study[51]. Further development of our results includes the demonstration of an atomic position-controlled method to generate and manipulate interlayer exciton emission on-demand, which is crucial for quantum information and simulators[52] based on 2D semiconductors.

## Methods

### Sample fabrication

A standard electron beam lithography (EBL) process was carried out on an FEI system (Helios 600 NanoLab, FEI) to fabricate the nanopillars. Negative e-beam HSQ (6%) resist was spin-coated on a Si/SiO₂ substrate at 4000 rpm to form a 150 nm thick resist film and baked on a hotplate at 150 °C for 5 min. The electron high tension (EHT) of the EBL system was set to be 30 kV with a beam current of 30 pA. The line dose used was 1.2 μC/cm. A 1 % NaOH solution was used as the developer, and DI water was used as the stopper. Nanopillars fabricated on a Si/SiO₂ substrate have diameters ranging from 100 to 300 nm, and a height of 130 nm (see Supplementary Information Fig. S1).

WSe₂ and MoSe₂ monolayers were mechanically exfoliated from bulk materials (2D Semi) and transferred onto SiO₂ nanopillars using a standard method. The WSe₂ and MoSe₂ monolayers were first prepared and characterized on PDMS and then transferred onto the nanopillar array using the transfer stage (Meta Inc.) with an alignment resolution of <1 μm and an angle controllability of <1°. The heterostructures were encapsulated between two thin layers of h-BN with a thickness of several nanometers. The whole sample was then annealed in a high vacuum (10⁻⁷ mBar) at 200 °C for 2 h.

### Photoluminescence spectroscopy

Low-temperature (4.2 K) photoluminescence spectroscopy was carried out in home-built spectroscopy setups loaded into a cryostat. (AttoDry 1000). All low-temperature PL spectra were obtained using a 100 × objective (NA = 0.8) with a spot size of about 500 nm in the cryostat at 4 K. PL spectra were collected with a high-resolution spectrometer (Princeton Instrument, PyLoN CCD). The gratings used

in the spectrometers were 300 g/mm and 1200 g/mm, both blazed at 750 nm. The excitation laser was an energy-tunable continuous-wave laser (M squared) coupled to a single-mode fibre. For lifetime experiments, a pulsed 726-nm diode laser (maximum repetition rate of 80 MHz with a full width at half maximum of <80 ps) was used to excite the sample. The intensity map was obtained by detecting the emission using a superconducting single-photon detector (Gifford-McMahon cycle SSPD from SCONTEL) while scanning the sample using a pair of high-resolution stages (ANSx150/LT). To filter the single sharp peak from the dense and discrete emission lines, high transparency gratings (IBSNE Photonics, FSTG-SNIR966-915) were used after the PL collection side from the sample and then sent to the spectrometer or avalanche photodiodes (APDs). A time-correlated photon counting card (Picoharp, PH300) was used to obtain the time-resolved PL data.

## Data availability

The datasets that support the findings of this study are available from the corresponding authors upon reasonable request.

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

## Acknowledgements

This work is supported by the Singapore National Research Foundation through QEP programme (No. NRF2021-QEP2-03-P01, No. NRF2021-QEP2-03-P10) and Singapore Ministry of Education (MOE2016-T3-1-006 (S)). J.Z. acknowledges support of the National Key Research and Development Program of China (Grant No. 2017YFA0303401), National Natural Science Foundation of China (Grant No.12074371), CAS Interdisciplinary Innovation Team, Strategic Priority Research Program of Chinese Academy of Sciences (Grant No. XDB28000000).

## Author contributions

H.C. and W.G. conceived the project. H.C., Q.T., X.C., Y.Z., D.C., Z.M., Z.H, Z.Z., and J.J.H.E. performed the measurements. H.C., Q.T., N.W., and Y.L. fabricated the devices. H.C., Q.T., J.-M.L., and A.R. analyzed the data. A.R. and J.-M.L. performed the theoretical analysis and simulation. A.R. and H.C. wrote the manuscript with inputs from all authors. Y.S., N.P., Y.M., X.W., X.L., J.Z., and W.G. supervise the project. All authors contributed to the discussion of the results.

## Competing interests

The authors declare no competing interests.
