## [Peer Review File · Nature Communications]

Reviewers' Comments:

Reviewer #1:

Remarks to the Author:

The authors studied the spectra and lifetimes of LIXs, showing that most of the LIX peaks in molybdenum diselenide (MoSe₂)/tungsten diselenide (WSe₂) heterobilayer on nanopillars come from donor-acceptor pair (DAP) exciton recombination. The interlayer DAP exciton (DAP IX) emission can account for more than 75% of the experimentally observed peak positions. By considering this coupling and the DAP exciton energy-lifetime relationship, a DAP IX dynamic model was proposed, which fits well with the experimentally observed LIX energy-lifetime correlation.

Although the topic is timely, the experimental results and explanation are interesting, the experiment evidence is still weak. Firstly, there are more than one hundred peaks observed from PL which partially explained as interlayer localized DAP IX by the authors. However, it is important to rule out the localized defect impact so that the interlayer localized DAP IX theory will be more necessary and convincing. Secondly, a systematic study of the relationship between the strain and the coupling between the band-edge IX and localized DAP IX is needed. It is critical to show the controlled strain will tune the coupling, enhancing the DAP IX PL in a controlled way. Above points makes me not recommend the manuscript for a publication in Nature Communications, at least for the current version.

Below are several detailed points/questions which might help to improve the manuscript.

1. The authors need to provide more evidence to show the sharp peaks in Fig. 2 a,b,c originate from localized DAP IX, not localized defect. The nature of localized DAP IX and localized defects are different, while the behaviors of the two might be similar. More control experiments are needed to show localized DAP mechanism is one of the origins of LIXs, i.e., some LIXs cannot be explained with only localized defects. Moreover, band structure calculation is needed to show the existence of localized DAP in the heterostructure.

2. The fitting in Fig. 2 e-g is not strong evidence as a 0.2 meV uncertainty is considered in supp section 1. Could the author explain more about the statement: 'an observed peak matches the DAP IX peaks if its energy is within ± 0.2 meV from the calculated DAP IX peaks.' The heterostructure should have tensile strain on nanopillar. Why is $\pm 1.5\%$ strain variation considered? Will the heterostructure have both tensile and compress strain on nanopillar?

3. The author mentioned "the IX potential landscape created by the nanopillar-induced strain increases the coupling between the band-edge IX and localized DAP IX, enhancing the localized DAP IX emission." I suggest the author to show more experimental evidence to prove the relation between the nanopillar-induced strain and the coupling between the band-edge IX and localized DAP IX. More experiments are needed to control the strain. At least, from Fig.1f and Extended Data Fig. 5, could the author consider summarizing all the information about each nanopillar size (radius, height), theoretical strain, PL intensity (or other experiment result), extracted coupling strength between the band-edge IX and localized DAP IX? It will be helpful if the author could show a direct correlation between strain and coupling strength between the band-edge IX and localized DAP IX. In addition, the PL spectra of flat area are recommended to add into supplementary as a reference.

4. The author suggests that 'an increase in power also causes a power-induced blue shift in IX energy due to dipole-dipole repulsion, which can have different magnitudes for localized DAP IX and band-edge IX'. Could the author show the fitting of Fig. 3d with different incident laser power? It should be able to observe the energy difference change between localized DAP IX and band-edge IX, from PL and decay rates vs. emission energy plot, if the DAP IX theory is valid.

5. The author suggests that 'an increase in power also causes a power-induced blue shift in IX energy due to dipole-dipole repulsion, which can have different magnitudes for localized DAP IX and band-edge IX'. In the flat region, did the author observed the coupling strength and energy shifted between localized DAP IX and band-edge IX, due to the laser power difference?

Is the localized DAP IX only observed on nanopillars, not on flat region of the heterostructure? Has the band-edge IX in flat region been observed and showing same property as the band-edge IX on nanopillar?

As the key information of the manuscript is related to the localized DAP IX and band-edge IX, it is necessary to show more evidence on properties of both localized DAP IX and band-edge IX from different area of the sample.

6. A typo: '(see inset in Fig. 1d), where DAP IX decays faster than band-edge IX.' I think it should

be Fig. 3d.

7. In Extended Data Fig. 1 and Extended Data Fig. 5, the patterns of the nanopillars are different. Are they from the same sample? Could author add more details to explain.

Reviewer #2:

Remarks to the Author:

In the manuscript titled "Interlayer donor-acceptor pair excitons in MoSe₂/WSe₂ moiré heterobilayer", the authors proposed a donor-acceptor pair mechanism for the localized interlayer excitons recombination induced multiple emission peaks in the MoSe₂/WSe₂ heterobilayer. They also conducted numerical simulation and sophisticated photoluminescence experiments to validate the proposed model. The manuscript has been well written and the topic is timely. The following questions and comments should be answered before it can be considered as a publication in this journal.

1. What is the nature of the donor and acceptor in the material? Are they impurities, vacancies or any other things? How many types of donor/acceptors are there in the heterobilayer samples, and which is responsible for the interlayer exatons? What is the density of them? I suggest the authors to clearly address these important issues, otherwise the readers could be puzzled.
2. Is the emission moiré influenced by the lattice size (and moiré potential strength)? Does the LIX emission show any moiré lattice size dependent behavior? The small moiré unit cell comparable to the critical donor-acceptor distance (~8 nm) would be interesting to verify the proposed mechanism. Since the moiré lattice could be easily tuned by varying the twist angles. Further experimental results on MoSe₂/WSe₂ moiré heterobilayer with different moiré superlattices (twist angles) would be necessary to validate the model.
3. It is reported that the nanopillars underneath the film could significantly enhance the PL due to the strain induced potential. Such large area strain might also significantly affect the moiré lattice and the moiré potential. Therefore, it would be necessary to clarify whether the moiré lattice and the moiré potential on the nanopillar remains the same as in the flat heterobilayer?
4. Is it possible to validate the proposed mechanism by directly measuring PL on the intact moiré lattice of heterobilayer samples without extra strain (and somehow random, for example the wrinkles in the Extended Data Fig. 1)? That would be much easier to compare the results with the previous works in the literature (Nature volume 567, pages 66–70 (2019); Nature volume 594, pages 46–50 (2021)). Here with the nanopillar, how do you rule out the possibility that the intensive luminescence with multiple peaks is mainly due to the (nanopillar) strain potential induced localization, while the moiré potential becomes a neglectable minor effect?
5. For the proposed potential application, what is the advantage of using the hetero-bilayer TMDC with LIX as quantum emitter, considering the random multi-peak nature in such system? Is it possible to select certain wavelength?
6. The statement in page 1 "The moiré superlattice, which arises from the difference in lattice constant or a nonzero twist angle 8-10, can increase electron-electron interactions 11,12." is a little ambiguous. Firstly, the strong correlation effect of electrons in the moiré superlattice of bi-layer 2D materials requires particular twist angles (so called magic angles) to guarantee the formation of dispersionless flat band, not any nonzero twist angles. Secondly, such a magic angle moiré superlattice does NOT increase the electron-electron interaction (the e-e interaction is the intrinsic property of electron), but quench the dynamic energy, by which way the Coulomb interactions dominate over kinetic energy and give rise to strong correlation. I suggest the authors to rephrase the ambiguous statement.

Reviewer #3:

Remarks to the Author:

The manuscript *Interlayer donor-acceptor pair excitons in MoSe₂/WSe₂ moiré heterobilayer* by Hongbing Cai *et al* reports a donor-acceptor pair (DAP) mechanism to explain the localized interlayer excitons (LIX) in MoSe₂/WSe₂ heterobilayers. The authors focus on twisted TMD sitting on nanopillars. They first show some exploratory results (schematics, microscope image, SHG, etc.) to illustrate the device and the DAP IX model. They then demonstrate a good matching between experimental and theoretical results regarding the PL spectra from the nanopillar locations. They further verify the model with energy- and time-resolved PL measurements.

Overall, this manuscript attempts to provide a possible mechanism of LIX formation in moiré heterostructures, which can be useful for the community. However, their results and analysis are not solid and need more clarifications, and I would only recommend for publication on *Nature Communications* if the following questions could be addressed by the authors properly:

1. The authors conduct their measurements on nanopillars. Transferring atomically flat 2D materials onto such nanopillars will likely damage the 2D materials and generate tons of defects, since they are tall (130nm according to authors) and not atomically flat. In fact, their SEM image in Extended Data Fig. 1 shows that the heterostructure is quite deformed. In Fig. 1f, they see nothing on flat regions, but very strong emission on nanopillars. To me, this is more like a “creation” of PL emission, rather than an “enhancement”. How do they know that such emission is not from defect states?
2. Eq. (1) is the core of their DAP model for predicting the PL peak position, where R_m is given by the lattice constant and interlayer distance based on Eq. (S1) in the supplemental. For small m (e.g., R_1 and R_2), the term with interlayer distance δ is dominant in Eq. (S1), since no chemical bonds are formed between the two van der Waals layers and $\delta \gg a_0$. However, the authors do not mention how they determine δ . Is it based on an independent theoretical/experimental work (e.g. a first-principle calculation)? Or they first leave it as an unknown parameter in the equation, match the measured data as good as they can, and then reversely find the best value of δ ? If latter, the model would only make sense if δ was reasonable. In any case, they need to clarify how they choose this parameter in the model.
3. The authors use $60^\circ \pm 1^\circ$ MoSe₂/WSe₂. Due to very small lattice mismatch, the moiré period is about 20nm. What’s the reason for not using $0^\circ \pm 1^\circ$ MoSe₂/WSe₂? It has the same moiré period but the moiré potential well depth might be different, and it would be interesting to see if the maximum donor-acceptor distance remains ~ 8 nm, as they claimed for the $60^\circ \pm 1^\circ$ case.
4. A minor thing: The colormap shown in Fig 1f and Extended Data Fig. 5 is the total PL intensity integrated over a certain range, rather than intensity at a specific energy. It’s better to call it “integrated PL intensity” to avoid confusion with other “PL intensity” in the text.

Reviewer #1 (Remarks to the Author)

The authors studied the spectra and lifetimes of LIXs, showing that most of the LIX peaks in molybdenum diselenide (MoSe₂)/tungsten diselenide (WSe₂) heterobilayer on nanopillars come from donor-acceptor pair (DAP) exciton recombination. The interlayer DAP exciton (DAP IX) emission can account for more than 75% of the experimentally observed peak positions. By considering this coupling and the DAP exciton energy-lifetime relationship, a DAP IX dynamic model was proposed, which fits well with the experimentally observed LIX energy-lifetime correlation.

Although the topic is timely, the experimental results and explanation are interesting, the experiment evidence is still weak. Firstly, there are more than one hundred peaks observed from PL which partially explained as interlay localized DAP IX by the authors. However, it is important to rule out the localized defect impact so that the interlay localized DAP IX theory will be more necessary and convincing. Secondly, a systematic study of the relationship between the strain and the coupling between the band-edge IX and localized DAP IX is needed. It is critical to show the controlled stain will tune the coupling, enhancing the DAP IX PL in a controlled way. Above points makes me not recommend the manuscript for a publication in Nature Communications, at least for the current version.

Below are several detailed points/questions which might help to improve the manuscript.

Reply:

We are glad the reviewer found the topic and results interesting and timely. We thank the reviewer for thoroughly reading our manuscript and giving many insightful suggestions to improve our manuscript further. Following the reviewer's valuable suggestions, we have done additional experiments and analysis, including:

- *) PL measurement on samples with different twist angles,
- *) PL and lifetime measurement for the emission from the flat area,
- *) PL intensity measurement from nanopillars with different parameter values,
- *) transmission electron microscopy (TEM) and band structure analysis to identify DAP IX structure, and
- *) simulation of strain on nanopillar

The results support our claim of DAP IX observation. In particular, the twist angle dependence cannot be explained using localized defect only, while it agrees with the DAP IX framework. Moreover, the relation between the PL intensity and nanopillar-induced strain shows that the strain controls the coupling between the band-edge and DAP IX. We address the reviewer's comments in detail below.

Comment 1

The authors need to provide more evidence to show the sharp peaks in Fig. 2 a,b,c originate from localized DAP IX, not localized defect. The nature of localized DAP IX and localized defects are different, while the behaviors of the two might be similar. More control experiments

are needed to show localized DAP mechanism is one of the origins of LIXs, i.e., some LIXs cannot be explained with only localized defects. Moreover, band structure calculation is needed to show the existence of localized DAP in the heterostructure.

Reply:

We thank the reviewer for pointing out the contribution of localized defects. We agree that defects play important roles in exciton emission, including the DAP IX. Defects can create donor and acceptor energy levels required for DAP IX formation. While a single defect (i.e., impurity, vacancy, or nanobubble) can also result in defect-bound IX, it is unlikely that most of the observed PL emission comes from defect-bound IX for the following reasons:

1. The number of peaks and the peak linewidth in our PL spectrum depends on the moiré superlattice, as shown by comparing the 0° , 60° , and $20(40)^\circ$ twisted samples (Fig. R1). In particular, the $20(40)^\circ$ twisted sample shows only broad peaks, while the other two samples show multiple sharp peaks. Such moiré superlattice dependence should not be observed if most PL peaks come from defect-bound IX emission¹.

On the other hand, this phenomenon can be well-explained using the DAP IX. In particular, due to the moiré potential barrier, the donor and acceptor should be localized within the same moiré cell. Unlike in the other two samples, the $20(40)^\circ$ twisted sample has a smaller moiré superlattice constant, resulting in less probability of DAP IX formation.

Figure R1 | Twist angle dependence of DAP IX PL emission. Three samples with twist angles 0° , $20(40)^\circ$, and 60° are compared. No pronounced sharp peaks were observed for the $20(40)^\circ$ sample. The PL spectra are normalized to the maximum intensity for each spectrum. This figure is included as Supplementary Information Fig. S6 in the revised manuscript.

2. The fast component decay rate, signifying the direct decay from localized IX to ground state, shows a dependence on the coupling between localized and band-edge IX. For a defect-bound IX, we do not expect such dependence. Instead, as in the usual spatially-localized exciton cases^{2,3}, we expect higher energy defect-bound IX to decay faster than the lower energy one due to the larger number of available decay channels.

On the other hand, this can be well explained with the DAP IX model. In particular, the fast decay rate in the case of DAP IX is proportional to the Bohr radius of the electron or hole (whichever is the largest one). Due to the coupling between DAP IX and band-edge IX, the effective Bohr radius becomes larger, leading to an increase in the electron-hole wavefunction overlap, which results in a faster decay rate as DAP IX energy gets closer to the band-edge IX energy.

To show the possible existence of localized DAP IX with an energy around 1.3 to 1.4 eV, we first characterize the defect in our sample using transmission electron microscopy (TEM). The TEM measurement result of suspended WSe_2 and MoSe_2 monolayers are shown in Fig. R2a and R2b, respectively. The possible impurities and vacancies, together with their approximated density, are shown in Fig. R2c. The combined density of V_{Se} and $V_{\text{Se}}V_{\text{Se}}$ in MoSe_2 is ~ 7 times the expected density of the moiré superlattice ($\sim 3 \times 10^{11} \text{ cm}^{-2}$ for twist angle of $\sim 1^\circ$) while the combined density of V_{W} and Se_{W} is comparable to moiré superlattice one.

Figure R2 | Defect characterization using TEM. TEM measurement of **a**, WSe_2 and **b**, MoSe_2 . The impurities/vacancies are labelled. **c**, Summary of defect type and defect density. This figure is included as Supplementary Information Fig. S7 in the revised manuscript.

We then consider two different sources of donor (acceptor) level:

1. Impurity/vacancy

Based on the TEM result, the donor can be V_{Se} and $V_{\text{Se}}V_{\text{Se}}$ in MoSe_2 , while the acceptor can be V_{W} and Se_{W} in WSe_2 . The band structure of these defects based on DFT calculation with HSE06 functional^{4,5} are shown in Fig. R3. The alignment between MoSe_2 and WSe_2 band is obtained by considering band-edge IX energy of 1.4 eV and IX binding energy of 0.32 eV⁶.

Figure R3 | Donor and acceptor energy levels. The number shown is the distance of the individual level from the WSe₂ valence band maximum (VBM). This figure is included as Supplementary Information Fig. S8 in the revised manuscript.

2. Moiré- and strain-induced localization

Due to moiré superlattice and external strain, the electron/hole experience can be localized in a single moiré site. Moreover, there can also be lattice reconstruction which can enhance the localization⁷. This localization creates energy levels near the conduction (valence) band, which can also act as donor/acceptor energy levels.

Considering the high density of defects in MoSe₂, it is reasonable to assign the donor part of DAP IX to the defects in MoSe₂. From the band energy diagram (Fig. R3), the energy levels of the V_{Se} and V_{Se}V_{Se} defects in MoSe₂ are located around 1.3 to 1.36 eV above the VBM, comparable to the value of E_0 (i.e., the minimum DAP IX energy) in our case. This value of E_0 indicates that the acceptor level should be close to the VBM. Such acceptor level can be the moiré/strain-localized energy level near the WSe₂ valence band. The discussion here has been included in the main text, Supplementary Information Sections 4 and 5 of the revised draft.

Comment 2

The fitting in Fig. 2 e-g is not strong evidence as a 0.2 meV uncertainty is considered in supp section 1. Could the author explain more about the statement: ‘an observed peak matches the DAP IX peaks if its energy is within ± 0.2 meV from the calculated DAP IX peaks.’ The heterostructure should have tensile strain on nanopillar. Why is $\pm 1.5\%$ strain variation considered? Will the heterostructure have both tensile and compress strain on nanopillar?

Reply:

We agree with the reviewer that the fitting result in Fig. 2e-g alone cannot be strong evidence of DAP IX. The experimental evidence of DAP IX comes from this fitting result together with the fitted energy-lifetime relationship and the twist angle dependence of the PL spectrum (Fig. R1).

Regarding the statement mentioned by the reviewer above, the uncertainty of ± 0.2 meV is obtained considering the reported strain variation due to moiré superlattice, which is around $\pm 1.5\%$ ⁸. Considering sample-to-sample variation, we argue that $\pm 1\%$ is a reasonable value for the strain variation and use strain variation, $\sigma_s = ?$ %, in our calculation below.

The uncertainty of the DAP IX peak position due to this strain variation can be derived as

$$\begin{aligned}\sigma_E &= |E(R) - E(R + \sigma_R)| \\ &= (E - E_0) \left| \frac{(\sigma_R/R)}{1 + (\sigma_R/R)} \right| \\ &= (E - E_0) |\sigma_s / (1 + \sigma_s)| \approx (E - E_0) |\sigma_s|,\end{aligned}\tag{R1}$$

where $E(R) = E_0 + \frac{\alpha}{R}$, E_0 is the minimum DAP IX energy, R is the distance between the donor and acceptor, $\alpha = \frac{e^2}{4\pi\epsilon}$, e is the electron charge, ϵ is the effective permittivity, and $\sigma_R = R\sigma_s$ is the uncertainty in the value of R . Eq. (R1) is the same as Eq. (S2) in Supplementary Information. As discussed in the Supplementary Information, using this equation, we obtained $\sigma_E \approx 0.2$ meV. Supplementary Information Section 1 has been revised to include the information here.

Regarding the nature of the strain, both compressive and tensile strains should be considered. Moiré superlattice and the nanopillar can contribute to the strain. The strain due to moiré superlattice can be compressive or tensile, depending on the location^{8,9}. The nanopillar changes the sample topography, resulting in additional tensile strain on the nanopillar location (Fig. R4b, see the answer to comment 3 for simulation details).

Comment 3

The author mentioned “the IX potential landscape created by the nanopillar-induced strain increases the coupling between the band-edge IX and localized DAP IX, enhancing the localized DAP IX emission.” I suggest the author to show more experimental evidence to prove the relation between the nanopillar-induced strain and the coupling between the band-edge IX and localized DAP IX. More experiments are needed to control the strain. At least, from Fig. 1f and Extended Data Fig. 5, could the author consider summarizing all the information about each nanopillar size (radius, height), theoretical strain, PL intensity (or other experiment result), extracted coupling strength between the band-edge IX and localized DAP IX? It will be helpful if the author could show a direct correlation between strain and coupling strength between the band-edge IX and localized DAP IX. In addition, the PL spectra of flat area are recommended to add into supplementary as a reference.

Reply:

We thank the reviewer for the suggestion to compare the nanopillar parameter with their PL emission. All nanopillars have the same height of ~ 130 nm, with the distance between nanopillars being $4 \mu\text{m}$. The pillar diameter, theoretical strain, the PL intensity (obtained from

the intensity map in Extended Data Fig. 5), and the coupling strength for each nanopillar are compiled in Table R1.

Excitation location	Diameter (nm)	Strain (%)	PL count (kcps)	Coupling strength, η_D^{exp} (cps.nm ⁻²)
Dot 1	180	0.95	45	1.39
Dot 2	200	0.91	58	1.45
Dot 3	140	1.07	45	2.30
Dot 4	100	1.23	15	1.50
Dot 5	100	1.23	16	1.60
Dot 6	120	1.15	20	1.39
Dot 7	160	1.01	26	1.02
Dot 8	120	1.15	38	2.64
Dot 9	160	1.01	12	0.47
Dot 10	100	1.23	24	2.40
Dot 11	120	1.15	32	2.22
Dot 12	80	1.34	20	3.13
Flat area	500 (beam diameter)	0.00	10	≥ 0.04

Table R1 | The PL behaviour of DAP IXs on nanopillars array with different diameters. This table is included as Supplementary Information Table S1 in the revised manuscript.

The theoretical strain is obtained from the numerical simulation of a membrane deposited on a SiO₂ nanopillar substrate. Considering there is a 10-nm thick hBN between the thin TMD heterostructure and the SiO₂ substrate, the membrane is modelled as 10 nm hBN membrane (Young's modulus, $Y = 0.8$ TPa ; Poisson's ratio, $\nu = 0.2$; mass density, $\rho = 2100$ kg.m⁻³) in the COMSOL plate model. The van der Waals interaction between the membrane and the substrate is modelled using Lennard-Jones (LJ 6-12) potential¹⁰

$$V(r) = \Gamma_a \left(\left(\frac{d_e}{r} \right)^{12} - \left(\frac{d_e}{r} \right)^6 \right), \quad (\text{R2})$$

where r is the distance between the substrate surface and a point on the membrane, $\Gamma_a = 0.2$ J.m⁻² is the hBN-SiO₂ interfacial adhesive energy¹¹, and $d_e = 0.6$ nm is the equilibrium interlayer distance without strain. The nanopillar substrate exerts van der Waals force on the membrane in the direction normal to the substrate surface (see Fig. R4a). The solution to the plate model is obtained using COMSOL. The two-dimensional simulated strain profile for pillar diameter, $d = 200$ nm, is shown in Fig R4b, showing an expected circular rotational symmetry. The plot of strain vs the distance from the centre of the pillar for various pillar diameters is shown in Fig. R4c.

Considering the coupling between DAP and band-edge IX, the collected DAP IX PL count rate can be expressed as

$$I_{\text{PL}} = C_0 \eta_D d_{\text{int}}^2 N_{\text{IX}}, \quad (\text{R3})$$

where N_{IX} is the IX population within the excitation beam diameter, d_{int} is the diameter of the effective interaction area, η_D is the coupling efficiency between DAP and band-edge IX, and $C_0 = \pi \eta_{\text{coll}} k_r n_{\text{DAP}} / 4$ is taken to be constant where n_{DAP} is the number of donor-acceptor pair per area, k_r is the average radiative decay rate of DAP IX, and η_{coll} is the collection efficiency.

The value of d_{int} and N_{IX} depends on the band-edge IX transport. For excitation at the nanopillar, the IX transport is mainly affected by the funnelling, which depends on the strain gradient. The strain gradient is negligible within the nanopillar area, and it does not have a strong dependence on the pillar diameter outside the nanopillar area (see Fig. R4c). Hence, for excitation on the nanopillar, the band-edge IX population will be concentrated in the nanopillar area, i.e., $d_{\text{int}} \approx d$, and N_{IX} does not depend on the pillar diameter. The coupling strength for the excitation on nanopillar can then be expressed as

$$\eta_D = \frac{I_{\text{PL}}}{C_0 N_{\text{IX}} d_{\text{int}}^2} \propto \eta_D^{\text{exp}} = \frac{I_{\text{PL}}}{d^2}, \quad (\text{R4})$$

where η_D^{exp} is the experimentally derived coupling strength metric. The values of η_D^{exp} for the nanopillars calculated using Eq. (R4) are included as the coupling strength in Table R1.

For excitation on the flat area, the band edge IX transport is mainly determined by the IX-IX repulsion and disorder. In the absence of disorder (e.g., nanobubble), the effective interaction area will be the same as the excitation area. However, the disorder will reduce this value. On the other hand, the IX-IX repulsion spreads the band-edge IX population, resulting in a smaller N_{IX} than the one in the case of nanopillar. Considering these two factors, the coupling strength for the excitation on the flat area can be expressed as

$$\eta_D^{\text{exp}} > \frac{I_{\text{PL}}}{d_{\text{beam}}^2}, \quad (\text{R5})$$

where $d_{\text{beam}} \sim 500$ nm is the beam diameter. The lower boundary of η_D^{exp} for the flat area calculated using Eq. (R5) is included as the coupling strength in Table R1.

Figure R4d shows the plot of η_D^{exp} as a function of the strain (σ) on the pillar. We also included the lower boundary of η_D^{exp} in flat area (i.e., zero strain) in this figure. We observed that, generally, a higher strain results in higher coupling strength. This result is agreeable with the coupling mechanism shown in Fig. 1b. More specifically, this coupling mechanism will have a Lorentzian dependence on the energy difference between the band-edge and DAP IX. The band edge IX energy is linearly redshifted by strain¹². On the other hand, the defect-related energy levels are less sensitive to strain and only experience redshift at high strain¹³. As a result, the strain dependence of the coupling strength should have a Lorentzian shape at low strain and saturate at high strain. In Fig. R4d, we have included the Lorentzian fitting of the experimental data, showing a good fit.

Figure R4 | Strain dependence of coupling strength. **a**, Strain simulation setting. The van der Waals force direction acting on the 10 nm thick membrane is normal to the nanopillar substrate. **b**, Two-dimensional simulated strain profile. The trace of the strain tensor is plotted. The dashed line indicates the position of the edge of the nanopillar with a diameter of 200 nm. **c**, One-dimensional simulated strain profile for various pillars. Here, r is the distance from the centre of the pillar. The strain on the nanopillar area is relatively uniform, with higher strain for smaller pillars. **d**, Coupling strength vs strain on nanopillar. The coupling strength between the band-edge and DAP IX increases with increasing strain. The symbols are measurement results, and the line is the Lorentzian fitting. This figure is included as Supplementary Information Fig. S13 in the revised manuscript.

The comparison between the PL spectrum from the flat area and nanopillar area is shown in Fig. R5. The dense and narrow peaks are also observed in the PL spectrum from the flat area, albeit with a much smaller intensity than those in the nanopillar area. The weaker PL in the flat area can be attributed to weaker DAP-(band-edge) coupling due to smaller strain and/or smaller N_{IX} due to the absence of funnelling. The discussion here has been included in Supplementary Information Section 7 and Fig. S1 of the revised draft.

Figure R5 | Comparison between PL spectrum from nanopillar and flat area. The excitation power is ~ 40 nW. Dense and sharp peaks are observed in all spectra. The PL spectrum from the flat area is multiplied by 12.5 times. The distance between the major ticks is 250 cps. This figure is included in the revised manuscript as part of Supplementary Information Fig. S1.

Comment 4

The author suggests that ‘an increase in power also causes a power-induced blue shift in IX energy due to dipole-dipole repulsion, which can have different magnitudes for localized DAP IX and band-edge IX’. Could the author show the fitting of Fig. 3d with different incident laser power? It should be able to observe the energy difference change between localized DAP IX and band-edge IX, from PL and decay rates vs. emission energy plot, if the DAP IX theory is valid.

Reply:

We appreciate this valuable suggestion from the reviewer. The power dependence of the nanopillar (dot 3)’s lifetime (decay rate)-energy correlation is plotted in Fig. R6. As the reviewer mentioned, it is possible to observe the change in the energy difference between localized DAP IX and band-edge IX from the power-dependent lifetime-energy correlation.

The DAP IX peak position does not change much with power, indicating negligible power-induced blue shift for DAP IX. Such behaviour is reasonable, given the localized nature of DAP IX. Conversely, the band-edge IX (trion in Fig. R6) experiences a noticeable power-induced blueshift. Given that the decay rate is maximum for the DAP IX peak closest to band-edge IX, the energy position of the decay rate maximum will also be blueshifted, as observed in Fig. R6. The discussion here has been included in the main text and Supplementary Information Fig. S4 of the revised draft.

Figure R6 | Power dependence of lifetime-energy correlation of dot 3. The black arrows show the position of the band-edge trion peak. The symbols are measurement results, and the lines are theoretical calculations. This figure is included as Supplementary Information Fig. S4 in the revised manuscript.

Comment 5

The author suggests that ‘an increase in power also causes a power- induced blue shift in IX energy due to dipole-dipole repulsion, which can have different magnitudes for localized DAP IX and band-edge IX’. In the flat region, did the author observed the coupling strength and energy shifted between localized DAP IX and band-edge IX, due to the laser power difference? Is the localized DAP IX only observed on nanopillars, not on flat region of the heterostructure? Has the band-edge IX in flat region been observed and showing same property as the band-edge IX on nanopillar?

As the key information of the manuscript is related to the localized DAP IX and band-edge IX, it is necessary to show more evidence on properties of both localized DAP IX and band-edge IX from different area of the sample.

Reply:

We thank the reviewer for this insightful inquiry. We observed the dense and sharp peaks everywhere, i.e., not only in the nanopillar area but also in the flat area. In the previous draft, we focused on the nanopillar area because the PL is much brighter than in the other areas (see Fig. R5 for a comparison between the PL in the nanopillar and flat area). As the reviewer has suggested, we have characterized the power- and energy-dependence of the PL in the flat area and found that it has a similar behaviour as the one obtained from the nanopillar area, i.e.:

1. The positions of the sharp peaks do not change with power, while the overall PL strength (envelope) shows a blue shift (Fig. R7), indicating the coupling strength and energy shifted between DAP and band-edge IX.

Figure R7 | Power dependence of the PL from the flat area. The dashed lines indicate the tracking of some peaks. This figure is included as Supplementary Information Fig. S3 in the revised manuscript.

2. The decay rate vs peak energy plot shows an appearance of maxima near the band-edge IX energy (Fig. R8), indicating a similar coupling behaviour between band-edge and DAP IX in the nanopillar and flat areas.

Figure R8 | Energy dependence of the PL dynamics in the flat area. **a**, Energy- and time-resolved PL of LIX emission from the flat area. Data (shown as symbols) was obtained under 100 nW excitation. The lines are double exponential decay fitting results. The error bars are smaller than the symbol size. **b**, Decay rates vs emission energy. The lines are guides for the eye. Two maxima with energy separation of ~ 5 meV are detected, corresponding to band-edge neutral exciton and trion. This figure is included as Supplementary Information Fig. S2 in the revised manuscript.

The discussion here has been included in the main text, Supplementary Information Fig. S2, and Fig S3 of the revised draft.

Comment 6

A typo: '(see inset in Fig. 1d), where DAP IX decays faster than band-edge IX.' I think it should be Fig. 3d.

Reply:

We thank the reviewer for noticing this mistake. This typo has been corrected in the main text of the revised draft.

Comment 7

In Extended Data Fig. 1 and Extended Data Fig. 5, the patterns of the nanopillars are different. Are they from the same sample? Could author add more details to explain.

Reply:

We appreciate the reviewer's inquiry. The Extended Data Fig. 1 and Extended Data Fig. 5 are taken from the same sample, but from different areas (see Fig. R9). The discussion here has been included in the caption of Extended Data Fig. 1 and Extended Data Fig. 5, and Supplementary Information Fig. S5 of the revised draft.

Figure R9 | Alignment between different areas of the sample. The red box indicates the pattern set repeated over the sample during the fabrication. This figure is included as Supplementary Information Fig. S5 in the revised manuscript.

- 1 MahdikhanySarvejahany, F. *et al.* Localized interlayer excitons in MoSe₂-WSe₂ heterostructures without a moiré potential. *Nat. Commun.* **13**, 5354, doi:10.1038/s41467-022-33082-6 (2022).

- 2 Steinhoff, A. *et al.* Combined influence of Coulomb interaction and polarons on the
carrier dynamics in InGaAs quantum dots. *Phys. Rev. B* **88**, 205309,
doi:10.1103/PhysRevB.88.205309 (2013).
- 3 Choi, J. *et al.* Twist angle-dependent interlayer exciton lifetimes in van der Waals
heterostructures. *Phys. Rev. Lett.* **126**, 047401, doi:10.1103/PhysRevLett.126.047401
(2021).
- 4 Zheng, Y. J. *et al.* Point defects and localized excitons in 2D WSe₂. *ACS Nano* **13**,
6050-6059, doi:10.1021/acsnano.9b02316 (2019).
- 5 Akkoush, A., Litman, Y. & Rossi, M. A hybrid-DFT study of intrinsic point defects in
MX₂ (*M*=Mo, W; *X*=S, Se) monolayers. *Phys. Status Solidi A* **n/a**,
doi:https://doi.org/10.1002/pssa.202300180 (2023).
- 6 Torun, E., Miranda, H. P. C., Molina-Sánchez, A. & Wirtz, L. Interlayer and intralayer
excitons in MoS₂/WS₂ and MoSe₂/WSe₂ heterobilayers. *Phys. Rev. B* **97**, 245427,
doi:10.1103/PhysRevB.97.245427 (2018).
- 7 Li, E. *et al.* Lattice reconstruction induced multiple ultra-flat bands in twisted bilayer
WSe₂. *Nat. Commun.* **12**, 5601, doi:10.1038/s41467-021-25924-6 (2021).
- 8 Shabani, S. *et al.* Deep moiré potentials in twisted transition metal dichalcogenide
bilayers. *Nat. Phys.* **17**, 720-725, doi:10.1038/s41567-021-01174-7 (2021).
- 9 Rodríguez, Á., Varillas, J., Haider, G., Kalbáč, M. & Frank, O. Complex strain scapes
in reconstructed transition-metal dichalcogenide moiré superlattices. *ACS Nano* **17**,
7787-7796, doi:10.1021/acsnano.3c00609 (2023).
- 10 Li, H. *et al.* Optoelectronic crystal of artificial atoms in strain-textured molybdenum
disulphide. *Nat. Commun.* **6**, 7381, doi:10.1038/ncomms8381 (2015).
- 11 Rokni, H. & Lu, W. Direct measurements of interfacial adhesion in 2D materials and
van der Waals heterostructures in ambient air. *Nat. Commun.* **11**, 5607,
doi:10.1038/s41467-020-19411-7 (2020).
- 12 He, Y. *et al.* Strain-induced electronic structure changes in stacked van der Waals
heterostructures. *Nano Lett.* **16**, 3314-3320, doi:10.1021/acs.nanolett.6b00932 (2016).
- 13 Linhart, L. *et al.* Localized intervalley defect excitons as single-photon emitters in
WSe₂. *Phys. Rev. Lett.* **123**, 146401, doi:10.1103/PhysRevLett.123.146401 (2019).

Reviewer #2 (Remarks to the Author)

In the manuscript titled “Interlayer donor-acceptor pair excitons in MoSe₂/WSe₂ moiré heterobilayer”, the authors proposed a donor-acceptor pair mechanism for the localized interlayer excitons recombination induced multiple emission peaks in the MoSe₂/WSe₂ heterobilayer. They also conducted numerical simulation and sophisticated photoluminescence experiments to validate the proposed model. The manuscript has been well written and the topic is timely. The following questions and comments should be answered before it can be considered as a publication in this journal.

Reply:

We are glad that the reviewer found the manuscript to be well written and the topic is timely. We thank the reviewer for thoroughly reading our manuscript and for the valuable inputs related to the nature of DAP, the effect of strain and moiré superlattice, and the possible application of our findings. Following the reviewer’s comments, we have conducted more experiments and analyses to substantiate our findings, including:

- *) transmission electron microscopy (TEM) and band structure analysis to identify DAP IX structure,
- *) PL measurement on samples with different twist angles,
- *) electron diffraction analysis on flat and nanopillar areas to study the effect of strain on the lattice structure, and
- *) PL measurement and Raman spectroscopy on the flat and wrinkle areas.

The results support our claim of DAP IX observation. In particular, the sample with a small/negligible moiré superlattice does not show sharp peaks, which is agreeable with the DAP IX framework. We address the reviewer’s comments in detail below.

Comment 1

What is the nature of the donor and acceptor in the material? Are they impurities, vacancies or any other things? How many types of donor/acceptors are there in the heterobilayer samples, and which is responsible for the interlayer exatons? What is the density of them? I suggest the authors to clearly address these important issues, otherwise the readers could be puzzled.

Reply:

We thank the reviewer for asking about these important points. There are two possible origins of the localized donor and acceptor energy levels:

1. Impurity/vacancy

The impurity/vacancy can result in localized energy levels in the band gap.

2. Moiré- and strain-induced localization

Due to moiré superlattice and external strain, the electron/hole experience can be localized in a single moiré site. Moreover, there can also be lattice reconstruction which can enhance the localization⁷. This localization creates energy levels near the

conduction (valence) band, which can also act as donor/acceptor energy levels.

To identify which kind of donor/acceptor sources result in localized DAP IX with an energy of around 1.3 to 1.4 eV, we first characterize the defect in our sample using transmission electron microscopy (TEM). The TEM measurement result of WSe₂ and MoSe₂ monolayers are shown in Fig. R10a and R10b, respectively. The possible impurities and vacancies, together with their approximated density, are shown in Fig. R10c. The combined density of V_{Se} and V_{Se}V_{Se} in MoSe₂ is ~ 7 times the expected density of the moiré superlattice ($\sim 2 \times 10^{11} \text{ cm}^{-2}$), while the combined density of V_W and S_{ew} is comparable to moiré superlattice one.

Figure R10 | Defect characterization using TEM. TEM measurement of **a**, WSe₂ and **b**, MoSe₂. The impurities/vacancies are labelled. **c**, Summary of defect type and defect density. This figure is included as Supplementary Information Fig. S7 in the revised manuscript.

We then analyse the energy levels corresponding to these defects. The band structure of these defects based on DFT calculation with HSE06 functional^{4,5} are shown in Fig. R11. The alignment between MoSe₂ and WSe₂ band is obtained by considering band-edge IX energy of 1.4 eV and IX binding energy of 0.32 eV⁶. Considering the high density of defects in MoSe₂, it is reasonable to assign the donor part of DAP IX to the defects in MoSe₂. From the band energy diagram (Fig. R11), the energy levels of the V_{Se} and V_{Se}V_{Se} defects in MoSe₂ are located around 1.3 to 1.36 eV above the VBM, comparable to the value of E_0 (i.e., the minimum DAP IX energy) in our case. This value of E_0 indicates that the acceptor level should be close to the VBM. Such acceptor level can be the moiré/strain-localized energy level near the WSe₂ valence band. The discussion here has been included in the Supplementary Information Section 5 of the revised draft.

Figure R11 | Donor and acceptor energy levels. The number shown is the distance of the individual level from the WSe₂ valence band maximum (VBM). This figure is included as Supplementary Information Fig. S8 in the revised manuscript.

Comment 2

Is the emission moiré influenced by the lattice size (and moiré potential strength)? Does the LIX emission show any moiré lattice size dependent behavior? The small moiré unit cell comparable to the critical donor-acceptor distance (~ 8 nm) would be interesting to verify the proposed mechanism. Since the moiré lattice could be easily tuned by varying the twist angles. Further experimental results on MoSe₂/WSe₂ moiré heterobilayer with different moiré superlattices (twist angles) would be necessary to validate the model.

Reply:

We thank the reviewer for the question and insightful suggestion of comparing the PL in different moiré lattices. We have conducted PL measurements on samples with different twist angles, which results in different moiré superlattice sizes and potential strengths.

We found that the number of peaks and the peak linewidth in our PL spectrum depends on the moiré superlattice, as shown by comparing the 0°, 60°, and 20(40)° twisted samples (Fig. R12). In particular, the 20(40)° twisted sample shows only broad peaks, while the other two samples show multiple sharp peaks. Such moiré superlattice dependence can be well-explained using the DAP IX framework. In particular, due to the moiré potential barrier, the donor and acceptor should be localized within the same moiré cell. Unlike in the other two samples, the 20(40)° twisted sample has a smaller moiré lattice constant, resulting in less probability of DAP IX formation. The discussion here has been included in the main text and Supplementary Information Section 4 of the revised draft.

Figure R12 | Twist angle dependence of DAP IX PL emission. Three samples with twist angles 0° , $20(40)^\circ$, and 60° are compared. No pronounced sharp peaks were observed for the $20(40)^\circ$ sample. The PL spectra are normalized to the maximum intensity for each spectrum. This figure is included as Supplementary Information Fig. S6 in the revised manuscript.

Comment 3

It is reported that the nanopillars underneath the film could significantly enhance the PL due to the strain induced potential. Such large area strain might also significantly affect the moiré lattice and the moiré potential. Therefore, it would be necessary to clarify whether the moiré lattice and the moiré potential on the nanopillar remains the same as in the flat heterobilayer?

Reply:

We thank the reviewer for this valuable suggestion. To investigate the effect of nanopillar-induced strain on the sample structure, we have conducted TEM characterization on the flat and nanopillar area. For this purpose, we fabricate a new sample where the heterostructure sample sits on suspended nanopillars on hBN layer support (see Fig. R13), allowing the detection of electron transmission through the sample. The TEM and selected area electron diffraction (SAED) images from this heterostructure sample on nanopillar (diameter: 100-200 nm, height: ~ 130 nm) and flat areas are shown in Fig. R14 (upper and lower images correspond to areas around two different nanopillars). These images show that the twist angle in the nanopillar and flat areas can be quite different, while the monolayer lattice constant does not change much (see Fig. R15 for the linecut of the SAED images extracted from the upper images of Fig. R14).

However, we found that the nanopillar-induced strain can decrease or increase the moiré superlattice size. Hence, we cannot conclude that this size change is responsible for the PL enhancement. We can thus conclude that the PL intensity enhancement is mainly due to other strain-related effects instead of strain-induced moiré superlattice size modification. These effects can include (1) funnelling due to strain gradient and (2) strain-enhanced coupling between band-edge and DAP IX. The discussion here has been included in the Supplementary Information Section 6 of the revised draft.

Figure R13 | Heterostructure on pillar sample used in TEM measurement. **a**, Sample illustration. A standard TEM grid with a 200 nm low strain SiN membrane of a $100 \times 100 \mu\text{m}^2$ window with through-holes with diameters ranging from 4 to 20 μm created by FIB was used. The hBN support layer has a thickness of ~ 10 nm. A PMMA (A2) layer was spin-coated, and high-dose EBL was used to produce negative nanopillars with amorphous carbon. **b**, Sample image. This figure is included as Supplementary Information Fig. S10 in the revised manuscript.

Figure R14 | Effect of nanopillar-induced strain on moiré superlattice. Upper and lower images are images taken from areas around two different nanopillars. The twist angles and moiré superlattice constant (written in red) are shown. The strain on the nanopillar changes the moiré superlattice. This figure is included as Supplementary Information Fig. S11 in the revised manuscript.

Figure R15 | Linecut of SAED image of WSe₂ on nanopillar sample. The monolayer lattice constant does not change much between pillar and outside pillar areas. This figure is included as Supplementary Information Fig. S12 in the revised manuscript.

Comment 4

Is it possible to validate the proposed mechanism by directly measuring PL on the intact moiré lattice of heterobilayer samples without extra strain (and somehow random, for example the wrinkles in the Extended Data Fig. 1)? That would be much easier to compare the results with the previous works in the literature (Nature volume 567, pages 66–70 (2019); Nature volume 594, pages 46–50 (2021)). Here with the nanopillar, how do you rule out the possibility that the intensive luminescence with multiple peaks is mainly due to the (nanopillar) strain potential induced localization, while the moiré potential becomes a neglectable minor effect?

Reply:

We thank the reviewer for the valuable suggestions. Yes, we can directly measure the PL on the wrinkle and flat areas. As shown in Fig. R16, like in the nanopillar case, the PL signals from these areas also consist of dense and narrow peaks, albeit with weaker PL intensity. Such observation shows that the strain is not necessary for the formation of dense and narrow peaks. On the other hand, we do not observe these narrow peaks in the sample with a large twist angle (see Fig. R12), which shows that the moiré superlattice size is essential in the formation of these peaks.

Figure R16 | Comparison between PL spectrum from different areas on the same sample. The excitation power is ~ 40 nW. Dense and sharp peaks are observed in all spectra. The PL spectra from the wrinkle and flat areas are multiplied by 4 and 12.5 times, respectively. The distance between the major ticks is 250 cps. This figure is included as Supplementary Information Fig. S1 in the revised manuscript.

The role of strain is mainly to enhance the PL intensity through two mechanisms: (1) strain gradient-induced funnelling and (2) strain-enhanced coupling between the band-edge and DAP IX. This strain effect on PL intensity can also explain why the PL intensity from the wrinkle area is higher than that from the flat area. Compared to the Raman spectrum at the flat area, the Raman spectrum at the wrinkle area shows an apparent redshift of WSe_2 $A'+E'$ phonon mode¹⁴ (Fig. R17), indicating a larger strain and, thus, higher PL intensity. The discussion here has

been included in the main text, Supplementary Information Fig. S1, Sections 4 and 6 of the revised draft.

Figure R17 | Comparison between Raman spectrum in wrinkle and flat area. The symbols are measurement results, and the lines are guides for the eye. The WSe₂ A'+E' phonon mode in the wrinkle area is redshifted compared to that in the flat area, indicating a larger strain. This figure is included as Supplementary Information Fig. S9 in the revised manuscript.

Comment 5

For the proposed potential application, what is the advantage of using the hetero-bilayer TMDC with LIX as quantum emitter, considering the random multi-peak nature in such system? Is it possible to select certain wavelength?

Reply:

We agree with the reviewer that it is important to highlight the advantage of using hetero-bilayer TMDC with LIX as a quantum emitter. As the referee mentioned, the observed LIX emission has a multi-peak nature, which can be beneficial in terms of flexibility of emission wavelength choice. Using an optical filter, we can choose one wavelength from these peaks. When a smaller number of peaks are needed, it is possible to reduce them by using a smaller sample, as shown in a recent study¹⁵. Moreover, LIX has an out-of-plane dipole. Hence, its emission wavelength can be tuned precisely using an electric field. Considering all these aspects, LIX in the heterobilayer TMDC can be considered a versatile quantum emitter platform. The discussion here has been included in the main text of the revised draft.

Comment 6

The statement in page 1 “The moiré superlattice, which arises from the difference in lattice constant or a nonzero twist angle 8-10, can increase electron-electron interactions 11,12.” is a little ambiguous. Firstly, the strong correlation effect of electrons in the moiré superlattice of bi-layer 2D materials requires particular twist angles (so called magic angles) to guarantee the formation of dispersionless flat band, not any nonzero twist angles. Secondly, such a magic angle moiré superlattice does NOT increase the electron-electron interaction (the e-e interaction is the intrinsic property of electron), but quench the dynamic energy, by which way

the Coulomb interactions dominate over kinetic energy and give rise to strong correlation. I suggest the authors to rephrase the ambiguous statement.

Reply:

We thank the reviewer for this suggestion. The mentioned statement has been rewritten in the main text of the revised draft to:

“The moiré superlattice, which arises from the difference in lattice constant or a particular range of twist angle ⁸⁻¹⁰, can increase the effect of electron-electron interactions ^{11,12}.”

- 4 Zheng, Y. J. *et al.* Point defects and localized excitons in 2D WSe₂. *ACS Nano* **13**, 6050-6059, doi:10.1021/acsnano.9b02316 (2019).
- 5 Akkoush, A., Litman, Y. & Rossi, M. A hybrid-DFT study of intrinsic point defects in MX₂ (M=Mo, W; X=S, Se) monolayers. *Phys. Status Solidi A* **n/a**, doi:https://doi.org/10.1002/pssa.202300180 (2023).
- 6 Torun, E., Miranda, H. P. C., Molina-Sánchez, A. & Wirtz, L. Interlayer and intralayer excitons in MoS₂/WS₂ and MoSe₂/WSe₂ heterobilayers. *Phys. Rev. B* **97**, 245427, doi:10.1103/PhysRevB.97.245427 (2018).
- 7 Li, E. *et al.* Lattice reconstruction induced multiple ultra-flat bands in twisted bilayer WSe₂. *Nat. Commun.* **12**, 5601, doi:10.1038/s41467-021-25924-6 (2021).
- 14 Dadgar, A. M. *et al.* Strain engineering and Raman spectroscopy of monolayer transition metal dichalcogenides. *Chem. Mater.* **30**, 5148-5155, doi:10.1021/acs.chemmater.8b01672 (2018).
- 15 Kim, H. *et al.* Dynamics of moire trion and its valley polarization in microfabricated WSe₂/MoSe₂ heterobilayer. *arXiv preprint arXiv:2301.11012* (2023).

Reviewer #3 (Remarks to the Author)

The manuscript *Interlayer donor-acceptor pair excitons in MoSe₂/WSe₂ moiré heterobilayer* by Hongbing Cai *et al* reports a donor-acceptor pair (DAP) mechanism to explain the localized interlayer excitons (LIX) in MoSe₂/WSe₂ heterobilayers. The authors focus on twisted TMD sitting on nanopillars. They first show some exploratory results (schematics, microscope image, SHG, etc.) to illustrate the device and the DAP IX model. They then demonstrate a good matching between experimental and theoretical results regarding the PL spectra from the nanopillar locations. They further verify the model with energy- and time-resolved PL measurements.

Overall, this manuscript attempts to provide a possible mechanism of LIX formation in moiré heterostructures, which can be useful for the community. However, their results and analysis are not solid and need more clarifications, and I would only recommend for publication on *Nature Communications* if the following questions could be addressed by the authors properly.

Reply:

We are glad that the reviewer found our result potentially useful for the community. We appreciate the reviewer's effort in thoroughly reading our manuscript and for the valuable comments related to PL enhancement, calculation detail, and the role of twist angle. Following the reviewer's inputs, we have conducted more experiments and analyses supporting our findings, including:

- *) PL measurement on the flat area,
- *) PL measurement on samples with different twist angles, and
- *) analysis of the maximum donor-acceptor distance for the near 0° twisted sample.

The results support our claim of DAP IX observation. In particular, the maximum donor-acceptor distance for the near 0° twisted sample is similar to that in the near 60° twisted sample, agreeable with the DAP IX framework. We address the reviewer's comments in detail below.

Comment 1

The authors conduct their measurements on nanopillars. Transferring atomically flat 2D materials onto such nanopillars will likely damage the 2D materials and generate tons of defects, since they are tall (130nm according to authors) and not atomically flat. In fact, their SEM image in Extended Data Fig. 1 shows that the heterostructure is quite deformed. In Fig. 1f, they see nothing on flat regions, but very strong emission on nanopillars. To me, this is more like a "creation" of PL emission, rather than an "enhancement". How do they know that such emission is not from defect states?

Reply:

We thank the reviewer for highlighting the difference between PL emission creation and enhancement. We check the PL spectrum from the flat area to confirm if the nanopillar creates or enhances the PL emission. As shown in Fig. R18, dense and narrow peaks are also observed in the PL spectrum from the flat area, albeit with much smaller intensity than those in the

nanopillar area. Such observation shows that the nanopillar enhances the PL intensity instead of creating the PL emission. The weaker PL in the flat area can be attributed to a weaker coupling between DAP and band-edge IX due to smaller strain and/or smaller N_x due to the absence of funnelling. The discussion here has been included in Supplementary Information Fig. S1 and Section 6 of the revised draft.

Figure R18 | Comparison between PL spectrum from nanopillar and flat area. The excitation power is ~ 40 nW. Dense and sharp peaks are observed in all spectra. The PL spectrum from the flat area is multiplied by 12.5 times. The distance between the major ticks is 250 cps. This figure is included in the revised manuscript as part of Supplementary Information Fig. S1.

Comment 2

Eq. (1) is the core of their DAP model for predicting the PL peak position, where R_m is given by the lattice constant and interlayer distance based on Eq. (S1) in the supplemental. For small m (e.g., R_1 and R_2), the term with interlayer distance δ is dominant in Eq. (S1), since no chemical bonds are formed between the two van der Waals layers and $\delta \gg a_0$. However, the authors do not mention how they determine δ . Is it based on an independent theoretical/experimental work (e.g. a first-principle calculation)? Or they first leave it as an unknown parameter in the equation, match the measured data as good as they can, and then reversely find the best value of δ ? If latter, the model would only make sense if δ was reasonable. In any case, they need to clarify how they choose this parameter in the model.

Reply:

We agree with the reviewer regarding the importance of Eq. (1), and we apologize for not discussing the interlayer distance, δ , in detail. We used $\delta = 0.7$ nm in our work based on the MoSe₂/WSe₂ interlayer distance reported in the literature^{1,16}. Supplementary Information Section 1 has been revised to include the information here.

Comment 3

The authors use $60^\circ \pm 1^\circ$ MoSe₂/WSe₂. Due to very small lattice mismatch, the moiré period is about 20nm. What's the reason for not using $0^\circ \pm 1^\circ$ MoSe₂/WSe₂? It has the same moiré period but the moiré potential well depth might be different, and it would be interesting to see

if the maximum donor- acceptor distance remains $\sim 8\text{nm}$, as they claimed for the $60^\circ \pm 1^\circ$ case.

Reply:

We thank the reviewer for the suggestion to check the MoSe₂/WSe₂ with a near 0° twist angle (i.e., AA stacked sample). We have measured the PL spectrum from the sample with near 0° twist angle and from a 20 (40) $^\circ$ twist angle sample (see Fig. R19). We found that the number of peaks and the peak linewidth in our PL spectrum depends on the moiré superlattice, as shown by comparing the 0° , 60° , and $20(40)^\circ$ twisted samples (Fig. R19). In particular, the $20(40)^\circ$ twisted sample shows only broad peaks, while the other two samples show multiple sharp peaks. Such moiré superlattice dependence can be well-explained using the DAP IX framework. In particular, due to the moiré potential barrier, the donor and acceptor should be localized within the same moiré cell. Unlike in the other two samples, the $20(40)^\circ$ twisted sample has a smaller moiré lattice constant, resulting in less probability of DAP IX formation.

The maximum donor-acceptor distance of the near 0° twisted sample can be obtained following the same procedure done for the near 60° twisted sample. In particular, we first find the E_0 by considering that the distance between two adjacent separable Lorentzian peaks should be larger than 60% of their full-width-at-half-maximum (FWHM $\sim 1\text{ meV}$ for the near 0° twisted sample). Using this criterion, we obtain $E_0 \sim 1.303\text{ eV}$. By using this E_0 and the minimum energy of the detected DAP IX peak ($\sim 1.327\text{ eV}$) in Eq. (1) in the main text, we obtain a maximum donor-acceptor distance of $\sim 8\text{ nm}$. This value is similar to the near 60° twisted sample case, which is expected given that the moiré superlattice constants in these two cases are similar. The discussion here has been included in the main text and Supplementary Information Section 4 of the revised draft.

Figure R19 | Twist angle dependence of DAP IX PL emission. Three samples with twist angles 0° , $20(40)^\circ$, and 60° are compared. No pronounced sharp peaks were observed for the $20(40)^\circ$ sample. The PL spectra are normalized to the maximum intensity for each spectrum. This figure is included as Supplementary Information Fig. S6 in the revised manuscript.

Comment 4

A minor thing: The colormap shown in Fig 1f and Extended Data Fig. 5 is the total PL intensity integrated over a certain range, rather than intensity at a specific energy. It's better to call it "integrated PL intensity" to avoid confusion with other "PL intensity" in the text.

Reply:

We thank the reviewer for this suggestion. The captions of Fig. 1f and Extended Data Fig. 5 have been updated in the revised draft accordingly.

- 1 MahdikhanySarvejahany, F. *et al.* Localized interlayer excitons in MoSe₂-WSe₂ heterostructures without a moiré potential. *Nat. Commun.* **13**, 5354, doi:10.1038/s41467-022-33082-6 (2022).
- 16 Li, W., Lu, X., Dubey, S., Devenica, L. & Srivastava, A. Dipolar interactions between localized interlayer excitons in van der Waals heterostructures. *Nat. Mater.* **19**, 624-629, doi:10.1038/s41563-020-0661-4 (2020).

Reviewers' Comments:

Reviewer #1:

Remarks to the Author:

In this revision, the authors satisfactorily addressed all my technical remarks and concerns as well as the theoretical interpretations. The added experiment and discussion have improved the overall quality of the manuscript. The theory part of this work is approximation to support the experiment, the experiment result is solid. The authors proposed a donor-acceptor pair mechanism for the localized interlayer excitons recombination induced multiple emission peaks in the MoSe₂/WSe₂ heterobilayer, which is a valuable idea and insightful view in explaining the nature of the excitonic states in heterobilayers.

In view of comments from the other referee and corresponding authors' rebuttal, I would recommend this paper to be published on Nature Communications.

Reviewer #2:

Remarks to the Author:

In the revised version of the manuscript entitled " Interlayer donor-acceptor pair excitons in MoSe₂/WSe₂ moiré heterobilayer " by Hongbing Cai et. al., the authors has provided additional experimental (TEM, PL ect.) and theoretical results in response to the questions raised by the referees. These results have resolved most of my concerns except one issue. Please clearly address the significance of this work specifically in the application of " highly tunable single-photon sources ", since it is a motivation of the work (see first sentence of the paper).

I would recommend for publication if that point could be addressed properly.

PS. I'd like to apologize to the editorial team and the authors for the delay of my report. It was due to some unexpected personal issues.

Reviewer #3:

Remarks to the Author:

The authors added massive additional experiments and analysis. However, it is still not convincing to me how they ruled out the impact from localized defects based on their arguments. I still think defects play a major role in the PL emission on nanopillars. Therefore, I do not recommend publication in Nature Communications.

REVIEWERS' COMMENTS

Reviewer #1 (Remarks to the Author):

In this revision, the authors satisfactorily addressed all my technical remarks and concerns as well as the theoretical interpretations. The added experiment and discussion have improved the overall quality of the manuscript. The theory part of this work is approximation to support the experiment, the experiment result is solid. The authors proposed a donor-acceptor pair mechanism for the localized interlayer excitons recombination induced multiple emission peaks in the MoSe₂/WSe₂ heterobilayer, which is a valuable idea and insightful view in explaining the nature of the excitonic states in heterobilayers.

In view of comments from the other referee and corresponding authors' rebuttal, I would recommend this paper to be published on Nature Communications.

Reply:

We thank Reviewer #1 for his/her review and support for the publication of our work.

Reviewer #2 (Remarks to the Author):

In the revised version of the manuscript entitled " Interlayer donor-acceptor pair excitons in MoSe₂/WSe₂ moiré heterobilayer " by Hongbing Cai et. al., the authors has provided additional experimental (TEM, PL ect.) and theoretical results in response to the questions raised by the referees. These results have resolved most of my concerns except one issue. Please clearly address the significance of this work specifically in the application of " highly tunable single-photon sources ", since it is a motivation of the work (see first sentence of the paper).

I would recommend for publication if that point could be addressed properly.

PS. I'd like to apologize to the editorial team and the authors for the delay of my report. It was due to some unexpected personal issues.

Reply:

We thank Reviewer #2 for his/her review and support for the publication of our work. The " highly tunable single-photon sources " comes from the nature of the formation of LIX that holds nonzero out-of-plane electrical dipole, which can be tuned by vertical electric field as reported by Ref. 28. We have included the following sentence in the Discussion part of the manuscript:

“In terms of application, the multippeak nature and the electric field tunability of the LIX emission (i.e., due to its out-of-plane dipole direction) show that it is a versatile quantum emitter platform.”

Reviewer #3 (Remarks to the Author):

The authors added massive additional experiments and analysis. However, it is still not convincing to me how they ruled out the impact from localized defects based on their arguments. I still think defects play a major role in the PL emission on nanopillars. Therefore, I do not recommend publication in Nature Communications.

Reply:

We agree with the reviewer that the defects contribute to the PL emission on nanopillars, including the formation of DAP IX with dense and sharp peaks emission. We believe this mechanism plays a major role in the emission behaviour from the MoSe₂/WSe₂ heterobilayer. We have also stated in the manuscript that other mechanisms could take place besides the DAP IX mechanism. We include the following sentence in the Discussion part of the manuscript:

“In summary, we have theoretically proposed and experimentally confirmed a novel DAP IX mechanism as the origin of the dense and sharp IX emissions from MoSe₂/WSe₂ on nanopillars, besides other mechanisms that could take place.”